# A qualitative study of lived experiences of underrepresented electrical workers using creative non-fiction

Zhiyang Shi [1], Donia Obeidat[1,2,3‡], Ali Bani-Fatemi[1‡], Aaron Howe[1‡], Behdin Nowrouzi-Kia[1,4,5,6] *

1 Restore Lab, Department of Occupational Science and Occupational Therapy, Temerty Faculty of Medicine, University of Toronto, Toronto, Ontario, Canada, 2 Rehabilitation Science Institute, Temerty Faculty of Medicine, University of Toronto, Toronto, Ontario, Canada, 3 Department of Rehabilitation Sciences, Faculty of Applied Medical Sciences, Jordan University of Science and Technology, Irbid, Jordan, 4 Krembil Research Institute-University Health Network, Toronto, Ontario, Canada, 5 Institute for Mental Health Policy Research, Centre for Addiction and Mental Health, Toronto, Ontario, Canada, 6 Centre for Research in Occupational Safety & Health, Laurentian University, Sudbury, Ontario, Canada

☉ These authors contributed equally to this work.
‡ These authors also contributed equally to this work.
* behdin.nowrouzi.kia@utoronto.ca

## Abstract

### Objectives

Women, Indigenous peoples, racialized individuals, and persons with disabilities remain underrepresented in the electrical industry. This study explored the lived experience of underrepresented electrical workers related to their mental health and workplace integration.

### Methods

A qualitative narrative design was employed. One-on-one interviews were conducted with eleven participants who self-identified as women, Indigenous peoples, racialized individuals, and/or persons with disabilities. Interview data were first analyzed using a narrative thematic approach and informed the development of creative non-fictional stories.

### Results

Three stories were developed. Story 1 – "Asking for a ride: being a women electrician" illustrated the experiences of a woman apprentice who faced inadequate job site accommodations and sexism challenges in the workplace. Story 2 – "The lunch talk: Indigenous people and racialized individuals in the trade" highlighted the experiences of Indigenous and racialized participants who encountered language barriers and discriminatory comments. Story 3 – "Luke's notes: living and working with disability"

**Data availability statement:** Data cannot be shared publicly or deposited within public data repository because of the privacy and confidentiality policy of the University of Toronto Research Ethics Board (contact via: ethics. review@utoronto.ca). In accordance with the University of Toronto Research Ethics Board mandate, the corresponding author (behdin. nowrouzi.kia@utoronto.ca) is the designated institutional custodian of the data. Data access can be requested from the corresponding author via email (behdin.nowrouzi.kia@utoronto.ca) for researchers who meet the criteria for access to confidential data. The University of Toronto's Research Ethics Board is an institutional body independent of the corresponding author and serves as the durable point of contact for additional inquiries. Long-term data storage will also be maintained through the University of Toronto Research Ethics Board for ongoing data availability. The data are stored on secure, password-protected University of Toronto institutional servers.

**Funding:** ZS received financial support from the Ontario Electrical League (https://www. oel.org/). Grant number is not applicable. The Ontario Electrical League contributed to the study conceptualization, recruitment, and data collection.

**Competing interests:** The authors have declared that no competing interests exist.

demonstrated the impacts of physical disability on the daily work of electrical workers, particularly in managing the physical demands and mental health strains.

## Conclusions

Electrical workers from underrepresented groups experience persistent barriers to mental health and workplace integration, including a lack of accommodations, limited social support, and experiences of discrimination at the workplace. These individuals also reported challenges in seeking workplace support due to a "toughness" culture within the industry. Electrical employers should foster an inclusive organizational culture that prioritizes the health and psychosocial well-being of underrepresented workers.

## Introduction

Equity-deserving groups including women, Indigenous peoples, racialized individuals, and persons with disabilities are underrepresented in the Canadian skilled trades. Despite representing 50.6% of the Canadian population, women make up merely 7.4% of the skilled trade workforce [1]. Indigenous peoples account for 0.7% of the workforce (vs. 5% of the general population) [2], racialized individuals comprise 21% (vs. 26.5%) [3], and persons with disabilities represent 6.7% (vs. 27%) [4]. There remains a persistent gap in these individuals' representation and integration in the workplace. In the electrical sector, diversity appears to be even lower [3].

Several factors influence workplace integration in skilled trades. For example, social relationships with coworkers and employers play an important role in workplace engagement and job satisfaction [5]. Tradeswomen and racialized tradespeople report lacking workplace support and encountering challenges such as job insecurity [6–9]. The lived experiences of Indigenous workers and persons with disabilities are particularly underresearched, despite the importance of these communities to the Canadian labour force.

Electrical work presents a distinct context within the skilled trades due to its intensive physical and cognitive demands. Compared to many other trades (e.g., carpenter), electrical work has been characterized as the most cognitively demanding trade due to its sustained vigilance and constant exposure to hazardous work environments, such as electrical shocks [10]. As a result, electrical workers frequently report high levels of job-related stress [11], chronic fear, and poor mental health outcomes [6,12–14]. Despite these concerns, existing research has primarily focused on white, male electrical workers, leaving a significant gap in understanding the experiences of women, Indigenous peoples, racialized individuals, and persons with disabilities. Given that sexism, racism, and ableism may intensify these physical and cognitive demands [15–17], there is a need to investigate how underrepresented electrical workers navigate the intersection of these challenges [18].

The purpose of this study was to explore the lived experience of electrical workers who self-identify as women, Indigenous peoples, racialized individuals, and/or persons with disabilities, with particular attention to mental health and workplace integration.

The research questions were: 1) What are the workplace experiences of electrical workers from underrepresented groups? and 2) What suggestions do these individuals have for promoting mental health and workplace integration?

## Methods

### Design and philosophical paradigm

This study employed a narrative design, which focuses on constructing meaning from human experiences through storytelling [19]. The approach aligned with our research objective to offer insight into individuals' identities, values, and lived realities [20].

Our philosophical paradigm was interpretivist, meaning that we believed "reality is dependent on those interpreting it", and meanings were generated "through transactions between researcher and participant, each bringing their prior experiences and understandings to bear on the topic of investigation" [21]. The research team comprised both university-based investigators with expertise in occupational health, psychology, and disability studies, and community partners from the electrical trades.

### Participants recruitment

The study population included electrical apprentices, workers, and employers who self-identified as women, Indigenous peoples, racialized individuals, and/or persons with disabilities. Recruitment was conducted through purposive sampling in collaboration with the Ontario Electrical League, a provincial advocacy organization representing 38,000 non-unionized electrical workers. Eligible participants were initially contacted by the Ontario Electrical League staff via email and phone call to gauge their interest. With participants' consent, the Ontario Electrical League shared contact information (participants' email and/or phone number) with the research team. Response rates were not available because the Ontario Electrical League did not disclose the total number of participants contacted. Additional recruitment was carried out at the Ontario Electrical League's chapter meetings across Ontario. The research team attended and presented the research at the meetings. Participants contacted the research team by email following the meetings. Written consent was obtained prior to data collection. Recruitment took place between November 15, 2024 and March 31, 2025. All procedures were approved by the University of Toronto Research Ethics Boards.

### Data collection

First author (ZS) conducted one-on-one, semi-structured interviews with participants who provided informed consent. Each interview was audio-recorded and set to be within 30 minutes. In cases where the full interview guide (Appendix A) was not covered in the initial session, a second interview was conducted. Two shorter interviews allowed for greater rapport-building, improved data richness, and respected participants' time, while enhancing reflexivity and trust [22]. The interview guide was structured into three thematic sections to align with the research questions. The first section focused on building rapport with participants, understanding their backgrounds, and discussing their perspectives on the current workplace culture through questions such as "Can you tell me about yourself?" and "How would you describe the general work culture within your organization?". The second and third sections aimed to directly address the research questions on participants' experiences and perspectives related to mental health and workplace integration, such as "have you experienced any mental health challenges associated with your work?". Interviews were conducted online or by phone, based on participant preference and availability.

### Data analysis

All interviews were transcribed and imported into NVivo 12 [23]. We started with a narrative thematic analysis to identify common ideas and threads across participants [24]. Following the methodological guideline [25], two researchers

independently reviewed all transcripts and generated initial codes from each interview. These codes were then discussed between the coders until they reached agreement. This iterative coding process led to the developing of themes and sub-themes through collaborative reflection and questioning.

We used the thematic findings to develop creative non-fictional stories representing participants' lived experiences [26]. Creative non-fiction is a qualitative analytical practice used to represent participants' lived experiences and researchers' interpretations through a story-telling form [24]. It has been accepted as a valid and trustworthy method in sociological and psychological research, supported by the literature that provides systematic methodological guidance [27]. This method has been used to examine the lived experiences of underrepresented individuals in diverse contexts, including disability research [28]. We first mapped themes and sub-themes onto narrative structure elements (story beginning, middle, and ending) to form story proposals. Participant quotes were embedded in the proposals to maintain a clear connection between source data and narrative form. Using a combination of participant's quotes and the researcher's narrative writing, stories were developed through employing the writing techniques recommended in the literature, such as establishing scenes, developing characters, and showcasing dialogues [29]. Scenario-based story plots were primarily constructed by the research team through creative writing to establish contextual coherence for each story. The research team reviewed the story proposals and drafts, provided feedback, and refined the stories.

We ensured the rigor of the analysis through several strategies aligned with qualitative research standards. First, our data collection and analysis methods were coherent with our narrative design and interpretivist paradigm [21]. Second, we followed the established guideline for narrative thematic analysis and recommendations for creative non-fictional writing. Third, the involvement of two independent coders in the narrative thematic analysis process allowed multiple perspectives and interpretations to be acknowledged via investigator triangulation [30]. Both coders are men, racial minorities, not living with disabilities, while they have differing levels of research expertise in mental health, occupational health, and disabilities studies. Fourth, the full research team developed and refined the stories, which acted as critical friends to ensure reflexivity [31]. The executive director of the Ontario Electrical League contributed to validating the stories as a community expert. Their input ensured the stories were credible and authentic, strengthening the trustworthiness of the findings [32].

## Results

Eleven participants completed the interviews, representing a diverse sample of electrical workers, To preserve confidentiality given the small sample size, only an overall summary of participants characteristics is reported. Participants ranged in age from 20 to 67 years old (Mean = 41.3), and included six men and five women. Eight were married or in common-law relationship, and three were single. Nine participants had completed college/university or higher education. On average, participants had worked as electricians for 9.7 years and reported working 37.7 hours per week.

Three creative non-fictional stories were developed based on the interview data, with each representing the experiences of women (story 1), Indigenous peoples and racialized individuals (story 2), and persons with disabilities (story 3). Participants quotes are presented in italicized text. Story 1 incorporates quotes from five participants, story 2 has quotes from six participants, and story 3 includes quotes from one single participant. Quotes are not labeled by participant identifiers to reduce identification risk. Additional quotes that did not fit in the stories are presented in Appendix B.

### Story 1: Asking for a ride: being a woman electrician

Context: Natalie is a young woman apprentice who just started the fourth year of their apprenticeship.

When Natalie steps onto the job site, it's freezing. Natalie slings her toolbelt and gets to work. An hour in, Natalie feels the familiar pinch in her bladder. She looks around, knowing what she won't find: no portable toilet, no site facilities of any kind. She mutters: "*Being in the trades, a lot of the time you're working in fields in the middle of nowhere. If the guys have to go pee, they just go in the middle of the field, whatever. But for a girl, it's very, very difficult.*"

The only option is the coffee shop five minutes down the road. Natalie *is not covered under the company's insurance to drive the work van, so she has to rely on other people to drive her.* And asking for this help never sits right with her because it means admitting she's different. And different, in the trades, so often means less than. "*In this job, it's mostly meant that the guys assume that I can't do something because I'm a woman,*" Natalie reflects, "*If I go to lift something heavy, they're like, 'oh, no let me help you', which is totally fine. It's nice that they do that, but sometimes I feel like I don't need their help to do some stuff.*"

However, the discomfort is getting worse. Natalie swallows her pride and walks up to one of the guy coworkers and asks: "Hey, mind driving me to the coffee shop?"

The guy replies: "Sure."

In the van, Natalie breaks the silence: "You know, *I think women can be really hard on themselves whenever they start to think about wanting to be in the trade. When I first started, I know the guys were joking, but some of them said that I was a lesbian just because I was an electrician, which I'm not and they just assume that.*"

Her coworker nods and responds: "I heard you. This job's never been an easy one. There was *one girl I went to school with, she must have been 24, skinny. She's not going to be lifting heavy cable, but she is smart, doing all the calculations for voltage drop. I relied on her help to get through many things. She was truly intelligent, and I think she would be great in those crawl spaces, and I hope she does get a job. But I can imagine older electricians looking at her and saying, 'oh she's just a young girl, what can she know?'*"

Natalie exhales and says: "You are absolutely right. *At my old job honestly, I was very uncomfortable a lot of the time. The guys there got along great, but it was mostly awkward for me because you go to work and then they would hit on me all day. My old job, my foreman asked me to move in with him. I had to switch crews.*"

Back on site, Natalie reflects on the conversation they had in the car: "I appreciate what he just did and said. *Who you work for and who you work with makes a huge difference. I've been pretty lucky in the fact that if anybody makes a sexist comment, it's usually one of my colleagues that'll step forward and be like, 'hey no, that's not OK', before I can even say or do anything.*"

She picks up her tools again. "*I really wish that this company would hire more women, because at lunch break, it's boy talk. Sometimes it'd be nice to have women to hang out with. I find just generally being a woman, I'm not as close with them, like there's a wall between us that will never be crossed.*" Natalie continues the thoughts, "*Some high schools have mechanic shops. If there was more of this, and there are more examples of people on television or in the celebrity world showing females in this type of trades, more women would be attracted and curious about it. If you don't see those examples that visibly, then you never think of that as your option.*"

**Story 2: The lunch talk: Indigenous people and racialized individuals in the trade**

Context: Barry is a racial minority and a new immigrant to Canada. Barry works for Mike, who is an Indigenous person and owns a contracting business.

The sun casts long shadows across the job site as the crew gathers around for lunch. Barry reaches for a sandwich and glances at his coworkers who are chatting easily in English. Laughter erupts at a joke, but Barry doesn't get it. He just smiles anyway.

Mike, the owner of the company, notices Barry sitting a bit apart. He steps over and asks Barry: "How's it going?"

Barry hesitates: "Good… I mean it just…" He pauses, searching for better words. "*Sometimes I still have a hard time explaining things to other guys. I just take it easy. It makes it easier to just write it… or to draw it down… what they need to do, instead of explaining it.*"

Mike smiles and responds: "The language, eh? That's a big one."

Barry stops chewing and says: "I also feel…outside."

Mike sees the frustrations from Barry's face. He leans closer and whispers to him: "I hear you. For me, it's not the language, but being Indigenous, being a visible minority, the feeling of being on the outside never goes away. *I've had one really negative experience with a very racist customer, and that made me feel pretty awful. He was pretty racist and an anti-Indigenous person. My boss, who knows I'm an Indigenous person, kind of deflected the guy and redirected his attention. I just kept doing my thing and tried to ignore the gentlemen. I don't think he was trying to be malicious, but I think he was a very old school, racist kind of dude. And it definitely was a shaking moment.*"

Barry nods and says: "Sorry to hear that."

Mike speaks slowly: "Thanks. Before I started my own business, *I would be chosen for this one thing with one of the journeymen, and other people will be doing other things. I always wondered 'why don't they want me doing things on that team?' And I would start to spin my wheels a little bit on that. Then I realized that the jobs were equally as important. I'm just working with this person versus that person.*"

The rest of the crew is picking back up. Mike talks to Barry: "*We all really benefit when we see diversity in any way. Diversity makes everyone feel more welcome. It benefits the whole company, homeowner and everyone. And that extrapolates up for so many different skills and experiences.*" Mike continues. "*There's been a lot of inclusiveness, not just in our trade, but for the other trades. Since now there's a lot of immigrants coming into the country, I don't know what religion they have, but people who wear turbans, some of the rules and the laws in the job site have been changed, because the hard hat does not fit over their head and their turban.*"

Mike claps Barry gently on the shoulder and says: "I'm glad to have you here. *I definitely try to make a point of hiring women and hiring people from underrepresented groups, because I've seen such value by being supported as one of those people in the trade.*"

Barry looks over at Mike, a quiet smile forming.

**Story 3: Luke's notes: living and working with disability**

Context: Luke is doing his second-year apprenticeship with a local contractor. He was diagnosed with a physical disability before becoming an electrical apprentice.

Bundles of wires lay across the job site. Luke takes a deep breath and pulls a notebook from his pocket. "Take list first", he reminds himself, *"Before I had all this [disability], my memory was on point. I'd be able to remember measurements. Now I have to write it down on a piece of paper or on my phone, which kind of takes a little bit more time for me."*

While the other apprentices are already running wire, Luke crouches beside the truck and carefully writes out his tasks. He tries to do it quickly but knows that his body works differently now. Luke mutters to himself: *"Through the disability, my bones hurt consistently all the time. It doesn't matter if I'm climbing a ladder or bending over to pick up a juice box or whatever. It's the same amount of pain. I can be standing there and my body's aching 24/7."*

Luke's mind drifts back to when he was applying for this job, *"When I went to all the interviews, I didn't have [disability] on my resume, but it was something that I mentioned to them [employers] through that interview towards the end. In every job site I go on, whoever is in charge, I let them know because it's a safety thing for myself, and it's a safety thing for them, right? I don't try to let my disability to get into the way of my work. But at the end of the day, it's something that my employers got to accept."* However, disclosing the disability was not an easy decision at that time. Luke thinks, *"To be honest, I was a little bit nervous that they [employers] would either look at me a different way or be like, 'oh, we don't want to have a guy that misses days.' Would I have a label put on me? Would my resume be tossed off to the side just because of the [disability]?"* His thoughts are interrupted as he wraps up his notes.

"Hey buddy, I need your help here!", a coworker calls out.

"What's up?", Luke replies and heads over. He's eager to support his coworkers – because he knows what it feels like to be supported. *When he told his bosses about his disability, they've always made sure that there's somebody on sites who knows CPR.* Luke remembers one specific moment, *"Last summer we were doing the fire alarm testing. I don't*

*feel comfortable with all the strobe lights because of the [disability]. I told the head electrician and asked if he could find something else for me to do while all this testing was going around. He asked me to stay outside and let me know when to come back in. I definitely feel supported, because I would have a seizure if I stayed in the building, hypothetically."* Thinking of that, Luke smiles. He feels ready for whatever the day brings.

## Discussion

This study explored the lived experience of electrical workers from underrepresented groups. The narratives captured how participants navigated challenges related to mental health and workplace integration.

Story 1 illustrated the experiences of a woman apprentice who faced inadequate job site accommodations and sexism challenges. The severe underrepresentation of women in the electrical trades is compounded by the absence of basic supports, such as access to clean toilets and properly sized tools. Women participants also discussed how their relationships with employers and coworkers influenced their intention to remain or leave the job. They emphasized the necessity of receiving respect from man coworkers and experiencing a sense of relatedness with women coworkers. Similar to other trades [33], women electricians were perceived as competent workers. Increasing the visibility of woman role models within the education system may help attract younger generations to pursue these careers.

Story 2 highlighted the experiences of Indigenous and racialized participants, who encountered language barriers and workplace racism. As Canada continues to welcome immigrants from non-English and non-French-speaking countries [34], language proficiency has become a critical determinant of professional success [35]. In this study, participants reported receiving little support from the employer. Given the collaboration nature of electrical work, employers need to offer support to workers who speak Indigenous or non-official languages, such as language training and employment counselling [36] Employers also have a responsibility to address racism and discrimination, as these issues continue to be prevalent across the skilled trades [37]. Although our findings suggested that overt incidents of discrimination were not frequent, they could still have meaningful impacts on participants' emotional and psychosocial well-being. In the current study, participants described their experiences using terms such as "pretty awful" and "shaking". However, the emotional effects of such incidents may extend beyond these immediate reactions and could lead to long-term consequences for the mental health and wellbeing [6].

Story 3 demonstrated the impacts of living with a physical disability on the daily work, particularly in managing the physical demands and mental health strains. Individuals with disabilities are at greater risk for mental health conditions [38], and the physical and cognitive work demands may further exacerbate these issues. For example, participants reported experiencing heightened stress at job sites where the conditions could trigger symptoms related to their disability. We found that the participants' decision to disclose was driven by their personal health and safety priority. However, such decisions can be particularly difficult for individuals who prioritize career development and fear potential negative consequences [39]. Canadian Apprenticeship Forum (2009) suggested that only about half of the skilled trade workers had disclosed their disability [40]. While trade-specific data remain limited, 35% of Canadian workers with disabilities never request accommodations from employers [41]. Inclusive workplace must encompass individual accommodations (e.g., modified work hours) and an organizational culture that prioritizes health, safety, and psychosocial well-being.

Across all three stories, participants described mental health challenges similar to those experienced by electrical workers more broadly, such as anxiety, stress, and burnout [7]. However, our findings suggested that underrepresented workers tended to rely on non-workplace support to cope with the challenges. For instance, when asked participants how they coped with mental health strains resulting from the workplace, the most frequent answer was support from family and friends. An effective social support system, however, should also include relationships at work, such as relationships with employers and with colleagues/peers [42]. Our team's prior research has identified multiple industry-wide barriers to workers' mental health and psychosocial wellbeing, including the societal stigma surrounding skilled trade professions, limited workplace equity and inclusiveness, and a lack of mental health resources and awareness [42,43]. This current

study further identified the presence of a "toughness" culture, which had discouraged open discussion about mental health or limited their intention to seek support from the workplace. This internal culture could intersect with the societal stigma and may hinder workers from thriving in their careers, particularly for those who are underrepresented individuals [43]. Together, these findings emphasized the urgent need for cultural change related to mental health promotion within the industry. Our recommendations include refining employer-employee hierarchy, developing workplace mental health resources and social support system, and initiatives to normalize mental health conversations.

## Limitations

Each narrative reflects the experiences of individuals from a specific identity group; however, we did not examine the intersection between multiple underrepresented identities. Future research should adopt intersectional frameworks, such as Gender-Based Analysis Plus (GBA+), to understand better how overlapping identities shape workplace experiences. In addition, all participants were non-unionized workers, whose access to resources and support may differ from unionized counterparts. It was also possible that these participants were more actively engaged with the Ontario Electrical League. Expanding recruitment beyond the Ontario Electrical League and identifying less-active members may provide a more comprehensive view of experiences across diverse work settings. Such expansion can also enrich the data and allow for the development of more detailed stories. The current stories were intentionally kept concise (approximately 500 words) to prioritize participants' direct quotes. Future studies may construct longer, more detailed stories with enriched details while maintaining engagement and accessibility for non-academic audiences.

## Conclusion

Electrical workers from underrepresented groups, including women, Indigenous peoples, racialized individuals, and persons with disabilities, experience persistent barriers to mental health and workplace integration. Our study identified multiple key barriers, including a lack of accommodations, limited social support, and experiences of discrimination at the workplace. A "toughness" culture within the electrical industry has discouraged these individuals to seek workplace support. Employers in the electrical industry should foster an inclusive organizational culture that prioritizes the health and psychosocial well-being of underrepresented workers.

## Acknowledgments

We would like to express our gratitude to all research participants, the collaborators, Beatrice Sharkey, Wendy Dobison and RoseMary MacVicar-Elliott from the Ontario Electrical League. Zhiyang Shi contributed to the conceptualization and design of the study, data collection, analysis, and interpretation of results, and the manuscript writing. Donia Obeidat contributed to the conceptualization and design of the study, data collection, analysis, and interpretation of results, and the manuscript writing. Ali Bani-Fatemi contributed to the conceptualization and design of the study, interpretation of results, and the manuscript writing. Aaron Howe contributed to the conceptualization and design of the study, and interpretation of results, and the manuscript writing. Behdin Nowrouzi-Kia contributed to the conceptualization and design of the study, data collection, analysis, and interpretation of results, and the manuscript writing. The data of this study are available from the corresponding author upon request. No AI was used at any stage during research development & design, data collection, manuscript preparation etc.

## Author contributions

**Conceptualization:** Zhiyang Shi, Donia Obeidat, Ali Bani-Fatemi, Aaron Howe, Behdin Nowrouzi-Kia.

**Data curation:** Zhiyang Shi, Donia Obeidat, Ali Bani-Fatemi, Aaron Howe.

**Formal analysis:** Zhiyang Shi, Donia Obeidat.

**Investigation:** Zhiyang Shi, Donia Obeidat, Ali Bani-Fatemi.

**Methodology:** Zhiyang Shi, Donia Obeidat, Ali Bani-Fatemi, Aaron Howe, Behdin Nowrouzi-Kia.

**Project administration:** Zhiyang Shi, Behdin Nowrouzi-Kia.

**Resources:** Ali Bani-Fatemi, Aaron Howe, Behdin Nowrouzi-Kia.

**Software:** Zhiyang Shi.

**Supervision:** Behdin Nowrouzi-Kia.

**Validation:** Ali Bani-Fatemi, Aaron Howe, Behdin Nowrouzi-Kia.

**Visualization:** Zhiyang Shi.

**Writing – original draft:** Zhiyang Shi.

**Writing – review & editing:** Zhiyang Shi, Donia Obeidat, Ali Bani-Fatemi, Aaron Howe, Behdin Nowrouzi-Kia.

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
