## [Decision Letter · Decision Letter 0]

15 Dec 2025

Dear Dr. Nowrouzi-Kia,

Thank you for submitting your manuscript to PLOS ONE. After careful consideration, we feel that it has merit but does not fully meet PLOS ONE’s publication criteria as it currently stands. Therefore, we invite you to submit a revised version of the manuscript that addresses the points raised during the review process.

Please kindly address each comment from reviewer one. The context influencing the selection of electrical workers and not any other worker needs justification and the provision of background to situate the study in other similar work in the gender inclusivity space.

We look forward to receiving your revised manuscript.

Kind regards,

Mwazvita TB Dalu, PhD

Academic Editor

PLOS One

Journal Requirements:

“ZS received financial support from the Ontario Electrical League (https://www.oel.org/). Grant number is not applicable. The Ontario Electrical League contributed to the study conceptualization, recruitment, and data collection.”

4. In this instance it seems there may be acceptable restrictions in place that prevent the public sharing of your minimal data. However, in line with our goal of ensuring long-term data availability to all interested researchers, PLOS’ Data Policy states that authors cannot be the sole named individuals responsible for ensuring data access (http://journals.plos.org/plosone/s/data-availability#loc-acceptable-data-sharing-methods ).

Reviewers' comments:

Reviewer's Responses to Questions

**Comments to the Author**

1. Is the manuscript technically sound, and do the data support the conclusions?

Reviewer #1: Partly

2. Has the statistical analysis been performed appropriately and rigorously?

Reviewer #1: N/A

3. Have the authors made all data underlying the findings in their manuscript fully available?

Reviewer #1: No

4. Is the manuscript presented in an intelligible fashion and written in standard English?

Reviewer #1: Yes

Reviewer #1: I think this is likely very interesting data with promise for the future. That said, the current work is in need of substantial revisions prior to publication. The majority of my comments are contained in the attached document. Broadly speaking, I feel the article needs to be substantially developed. This includes additional discussion of the existing literature, the methodology used, the contents of the interviews, and the narrative stories developed. I do not think this is an insurmountable task, and, hopefully is not one that will require additional interviews to generate additional data. That said, I also do not feel as if minor revisions would be sufficient. As the article stands, it feels like a loosely connected collection of quotes from interviewees, which are then used to support conclusions that are not fully supported by what is presented. It feels like a lot of the interview data is hidden from the reader, and the discussion and conclusions are presented in somewhat of a "trust me, this is supported by the interviews" kind of way. I am not doubting the interviews do support the conclusions, but as a reader I need to see it.

**Do you want your identity to be public for this peer review?** For information about this choice, including consent withdrawal, please see our Privacy Policy

Reviewer #1: No

---

## [Author Response · Author response to Decision Letter 1]

20 Jan 2026

We truly appreciate editor's and reviewer's feedback on the manuscript. We have considered the additional comments from the reviewer and made modifications to the manuscript. We have outlined our responses and edits in the revised manuscript and responses to reviewer.

---

## [Editor Report · Decision Letter 1]

8 Feb 2026

Dear Dr. Nowrouzi-Kia,

Thank you for submitting your manuscript to PLOS ONE. After careful consideration, we feel that it has merit but does not fully meet PLOS ONE’s publication criteria as it currently stands. Therefore, we invite you to submit a revised version of the manuscript that addresses the points raised during the review process.

We look forward to receiving your revised manuscript.

Kind regards,

Mwazvita TB Dalu, PhD

Academic Editor

PLOS One

Journal Requirements:

Additional Editor Comments:

Thank you for a much improved submission. Please kindly address these few comments before we finalise the decision.

Methods

Add more detail on recruitment. How were they approached? Which method of modification of sampling? What was the study population? How many participants were approached and what was the response rate?

Data collection

Thank you for the appendix, please add detail to the structure and selection of the interview questions, and the justification behind this choice as it relates to the study aim.

Data analysis

Much improved, thank you. Please add a critical science approach by adding a reflexive note on the positionality of the two researchers involved in the narrative writing to make readers aware of any positional biases or otherwise.

---

## [Author Response · Author response to Decision Letter 2]

11 Feb 2026

We greatly appreciate the reviewer’s feedback on the revision. We have considered their comments and revise the manuscript accordingly. We have outlined our revision in the file "Response to reviewers"/

---

## [Editor Report · Decision Letter 2]

9 Mar 2026

A qualitative study of lived experiences of underrepresented electrical workers using creative non-fiction

PONE-D-25-55981R2

Dear Dr. Nowrouzi-Kia,

We’re pleased to inform you that your manuscript has been judged scientifically suitable for publication and will be formally accepted for publication once it meets all outstanding technical requirements.

Kind regards,

Mwazvita TB Dalu, PhD

Academic Editor

PLOS One
---

## [Editor Report · Acceptance letter]

PONE-D-25-55981R2

PLOS One

Dear Dr. Nowrouzi-Kia,

I'm pleased to inform you that your manuscript has been deemed suitable for publication in PLOS One. Congratulations! Your manuscript is now being handed over to our production team.

Kind regards,

on behalf of

Dr. Mwazvita TB Dalu

Academic Editor

PLOS One